# Sublethal Effects of *Beauveria bassiana* Strain BEdy1 on the Development and Reproduction of the White-Backed Planthopper, *Sogatella furcifera* (Horváth) (Hemiptera: Delphacidae)

**DOI:** 10.3390/jof9010123

**Published:** 2023-01-16

**Authors:** Yongbo Xia, Siyuan Yu, Qunfang Yang, Jing Shang, Yi He, Fuyun Song, Qing Li, Chunxian Jiang

**Affiliations:** College of Agronomy, Sichuan Agricultural University, Chengdu 611130, China

**Keywords:** *Sogatella furcifera*, entomogenous fungi, biological control, sublethal effect, age-stage two-sex life table

## Abstract

Rice (*Oryza sativa*) is the most important food crop all over the world, while white-backed planthopper (WBPH), *Sogatella furcifera* (Horváth) (Hemiptera: Delphacidae) is an important pest causing rice yield reduction. The purpose of this study is to evaluate the sublethal effects of strain BEdy1 *Beauveria bassiana* (Bals.-Criv.) Vuill. (Hypocreales: Cordycipitaceae) on *S. furcifera* using the two-sex life table analytical method, compare the life tables of the F_0_ and F_1_ generations of WBPHs which were treated with sublethal concentrations (LC_10_, LC_25_) of *B. bassiana* BEdy1 with a control group. The results showed that the duration of the egg, 4th-instar and 5th-instar nymph, pre-adult, total pre-oviposition (TPOP) and mean generation time (***T***) for the LC_25_ treatment were significantly longer than those of the control and LC_10_ treatment. However, the duration of the adult, the longevity of male and female adults and the oviposition days of female adults for the LC_25_ treatment were significantly shortened. The fecundity of female adults, intrinsic rate of increase (***r***), net reproductive rate (***R*_0_**) and finite rate of increase (***λ***) for the LC_25_ treatment were significantly decreased compared with those of other treatments. The duration of the egg and pre-adult stage for the LC_10_ treatment were longer than those of the control group, but the population parameters showed no significant difference. Therefore, the LC_25_ of *B. bassiana* BEdy1 can inhibit the population growth of *S. furcifera*.

## 1. Introduction

Rice (*Oryza sativa*), one of the most important cereal crops and the main food source for more than a third of the global population, is cultivated in over 100 countries currently, and Asian nations account for 90% of rice production in the world [1,2]. The white-backed planthopper (WBPH), *Sogatella furcifera* Horváth (Homoptera: Delphacidae), is a typical r-strategy pest, one of the destructive long-distance migratory pests of rice in Asia, its permanent breeding areas are in the tropics, and it commonly migrates from the subtropic to temperate areas [3,4,5]. Due to the fact that both adults and nymphs suck phloem sap from rice plants, which causes rice plants to turn orange-yellow, reduces plant vigor and slows growth, under suitable environmental conditions, feeding of a large number of planthoppers causes drying of rice leaves, withering of tillers and results in plant death called “hopper-burn” [6,7,8]. Additionally, Southern rice black-streaked dwarf virus (SRBSDV), one of the most important rice pathogens, is transmitted by WBPHs with high efficiency to rice fields in the main rice-growing areas in Asia, such as China, northern Vietnam and Japan, resulting in a severe drop in rice production [9,10,11]. Thus, effectively blocking the spread of WBPHs is undoubtedly a promising manner to avoid the threat of SRBSDV [12].

At present, applying chemical insecticides is still the most common means of preventing and managing *S. furcifera*. The large-scale and irrational use of chemical pesticides against pests has led to various problems such as environmental contamination, food security, phytotoxicity, disruption of non-target organisms, pest resurgence and insecticide resistance [1,13,14]. Previous studies have shown that the field populations of WBPHs have developed high levels of resistance to 15 compounds worldwide [15]; the rapid increase of pesticide resistance in WBPH populations has led to an emphasis on the development of environmentally friendly management methods.

*Beauveria bassiana* (Bals.-Criv.) Vuill. (Hypocreales: Cordycipitaceae), a cosmopolitan and soil-borne fungus, is commercialized as a biological insecticide worldwide and it has exhibited great potential in controlling many pests [16], including rice planthoppers under laboratory and field conditions [17,18,19]. However, the insecticidal effect is not always desirable due to the low effecting rate of *B. bassiana* when applied in the field. Infection efficiency depends on the number of conidia, relative humidity, temperature and ultraviolet radiation, and exposure to fungistatic compounds on the phylloplane can reduce the efficacy of entomogenous fungi [20,21,22]; accordingly, the sublethal effect becomes the most significant mode of action of *B. bassiana* on pests.

Sublethal effects contain physiological and behavioral effects, including effects on neurophysiology, development, adult longevity, immunology, fecundity, sex ratio, feeding behavior, oviposition behavior and learning performance [23]. To date, the sublethal effects of chemical pesticides commonly used in rice fields in China, such as imidocloprid, buprofezin, sulfoxaflor, triflumezopyrim, nitenpyram and clothianidin, have been reported on *S. furcifera* [3,24,25,26,27,28]. The sublethal effects caused by fungal infection have been observed in a variety of insects and mites, but the potential sublethal effects of *B. bassiana* on *S. furcifera* are still unknown.

*Beauveria bassiana* BEdy1, isolated from *Ergania doriae yunnanus* Heller (Coleoptera: Curculionidae), has great potential for development and application in the future [29]. In this study, we applied the theory of age-stage two-sex life table and TWOSEX-MSChart Software to evaluate the sublethal effects of LC_10_ and LC_25_ doses of *B. bassiana* BEdy1 on the biological traits of *S. furcifera* in various stages under the F_0_ and F_1_ generations [30,31]. The results of this study not only provide directions for sustainable control of this important pest in rice-growing areas but also provide useful data for pest resistance management and integrated pest management (IPM).

## 2. Materials and Methods

### 2.1. Rearing of Insects

*Sogatella furcifera* was collected from a rice field in Hongyanba (105.4756° E, 28.1880° N), Xuyong County, Luzhou City, Sichuan Province, China, in 2020 and reared on rice (TN1) seedlings (about 15 cm high) without exposure to any pesticides for more than 15 generations in the intelligent artificial climate chamber (RXE-450, Ningbo Jiangnan Instrument Factory Co., Ltd., Ningbo, China) under 26 ± 1 °C, 70% ± 10% relative humidity and a photoperiod of 14 h:10 h (L:D).

All experiments were performed in the Agricultural Pest Laboratory at the College of Agronomy, Sichuan Agricultural University.

### 2.2. Culturing of Beauveria bassiana Strain BEdy1

The *Beauveria bassiana* strain BEdy1 used in the test was provided by Shang Jing’s research group, Phytopathology Laboratory, College of Agronomy, Sichuan Agricultural University. The GenBank accession number of this fungus on the NCBI database is MK345993. The test strain was cultured on PPDA (potato 200 g, dextrose 20 g, agar 20 g, peptone 10 g) in plastic petri dishes (90 mm diameter), wrapped with Parafilm and placed in darkness at 26 ± 1 °C. After 10–12 days, *B. bassiana* was harvested using a sterile blade, followed by the conidial suspension being placed inside a reagent bottle containing 0.05% Tween-80 Sterile water for later experiments.

### 2.3. Bioassay

Using the spray method, treatment of WBPH 3rd-instar nymphs with *B. bassiana* strain BEdy1 at five concentrations (1 × 10^5^, 1 × 10^6^, 1 × 10^7^, 1 × 10^8^ and 1 × 10^9^ spores/mL), with 0.05% Tween-80 Sterile water treatment served as a control. Firstly, we observed the molting of nymphs under the microscope every day and collected thirty healthy and uniform 3rd-instar nymphs with a suction trap as one group. For each group, 3 mL of the spore suspension was sprayed on the body wall of 30 3rd-instar nymphs with a hand-held spray. After spraying, they were transferred to a plastic cup (500 mL) containing 15 rice seedlings with 4–5 leaf stage leaves that were washed with ddH_2_O. At last, the death number was recorded after 96 h, and the individual nymphs were gently touched with a small brush, and the nymphs that were weak and unable to crawl normally served as the death record. There were three replicates for each treatment; toxicity data of the LC_10_ and LC_25_ values, confidence intervals at the 95% level and chi-square values were calculated using POLO Plus 2.0 statistical software (LeOra Software Inc., Petaluma, CA, USA).

### 2.4. Determination of Life Table Parameters of F_0_ and F_1_

The sublethal effects of *B. bassiana* on the life table parameters of *S. furcifera* were followed by Xiang’s and Ali’s approach with some modifications [24,25]. Firstly, about 300 pairs of WBPH adults were collected and transferred to a clean cage (60 cm × 40 cm × 30 cm), feeding on fresh rice seedlings. After 24 h of spawning, these rice seedlings containing eggs were transferred to another similar cage until the eggs developed into 3rd-instar nymphs, and these sensitive 3rd-instar nymphs were treated as F_0_ individuals. Based on the bioassay result, the sensitive populations of 3rd-instar nymphs were treated with LC_10_ and LC_25_ of *B. bassiana* (the treatment group) and 0.05% Tween-80 (the control group). After 96 h of treatment according to the method in Section 2.3, 150 surviving nymphs were randomly selected, and each was transferred to a flat-bottomed test tube (diameter × height: 20.0 mm × 145.0 mm) containing 1 fresh rice seedling. The first 100 nymphs were numbered after the formal test, and the last 50 nymphs were numbered for replacement. The rice seedlings were replaced every 3 days, and the condition and stage of the tested insects were observed and recorded daily. After the nymphs’ eclosion, the male and female adults were paired one by one and transferred to a flat-bottomed test tube containing one fresh rice seedling. If there was no pairing with it, the pairing was selected from the substitute flat-bottomed test adults. The rice seedlings in test tubes were replaced daily, and the number of eggs in the replaced rice seedlings was checked under the stereo microscope till the adults died. The longevity and fecundity of the female adults were recorded.

The offspring of F_0_ treated with *B. bassiana* BEdy1 and the control treatment were collected as the F_1_ generation. In the first step, 150 1st-instar nymphs were randomly selected from the F_0_ offspring as the experimental population. The first 100 nymphs were numbered and used for the formal test, and the last 50 nymphs were numbered and used for replacement. They were fed separately in a test tube containing a rice seedling to observe the stage and condition of WBPHs. When these nymphs became adults, they were paired up as described above. The rice seedlings in test tubes were replaced every day. We checked the number of eggs in the replaced rice seedlings until all the adults died. Eventually, biological parameters were recorded, including the development duration, longevity and fecundity.

### 2.5. Statistical Analysis

Analysis of the raw life table data was conducted using the age-stage two-sex life table theory [32,33]. The basic life-table parameters were analyzed using the TWOSEX-MSChart Software [31]. Fundamental life-table parameters including the age-stage survival rate (***s_xj_***), female age-specific fecundity (***f_xj_***), population age-specific survival rate (***l_x_***), population age-specific fecundity (***m_x_***), age-life expectancy (***e_xj_***), age-stage-specific reproductive value (***v_xj_***), intrinsic rate of increase (***r***), finite rate (***λ***), net reproductive rate (***R*_0_**) and mean generation time (***T***) were calculated [34,35].

***s_xj_***, ***f_x_***, ***l_x_***, ***m_x_***, ***e_xj_***, ***l_x_m_x_*** and ***v_xj_*** were plotted using Sigmaplot 14.0 (Systat Software, Inc., San Jose, CA, USA). The mean and standard error values (100,000 replications were used in the bootstrapping procedures) of the life table parameters were accurately estimated using the bootstrapping technique [36]. The significant difference between parameters was calculated using the paired bootstrap test in the TWOSEX-MSChart Software [31].

## 3. Results

### 3.1. Sublethal Concentrations of B. bassiana BEdy1 to S. furcifera

The results (Table 1) showed that the sublethal concentrations LC_10_ and LC_25_ of *B. bassiana* to the 3rd-instar nymphs of the WBPH at 96 h were 8.29 × 10^3^ spores/mL and 1.65 × 10^6^ spores/mL, respectively.

### 3.2. Sublethal Effects of B. bassiana BEdy1 on the *F_0_* Generation

The result (Table 2) showed that after the 3rd-instar nymphs of WBPH were treated with sublethal concentrations of *B. bassiana* BEdy1, there was no significant difference in the lifespan of the F_0_ generation adults compared with the control group. The average number of eggs laid by the females of the F_0_ generation treated with LC_25_ was significantly lower than that of the control and LC_10_ treatment groups (*p* < 0.05).

### 3.3. Sublethal Effects of B. bassiana BEdy1 on the Development and Reproduction of the *F_1_* Generation

The results showed that the F_1_ generation of WBPH treated with LC_10_ and LC_25_ of *B. bassiana* was able to complete its development (Table 3). The development periods of the eggs of the LC_10_ and LC_25_ treatment groups (6.51 days and 6.86 days) were significantly longer than those of the control (5.66 days), and the development times of the 1st-instar nymph showed no significant differences between all three treatments. The development times of the 2nd-instar nymph (2.20 days) and 3rd-instar nymph (2.66 days) of the LC_10_ treatment were significantly shortened (*p* < 0.05), when compared with the control group. The development times of 4th-instar nymph (2.95 days), 5th-instar nymph (3.97 days) and pre-adult (21.11 days) of the LC_25_ treatment were significantly longer than those of the control group (*p* < 0.05). The development times of adult (13.87 days) and the survival time of male and female adults (♀ 12.43 days, ♂ 15.30 days) of the LC_25_ treatment group were the shortest of the three treatments, but the total pre-oviposition (TPOP) (24.76 days) of LC_25_ was significantly longer than the control and LC_10_ treatment groups (*p* < 0.05). There was no significant difference in the adult pre-oviposition (APOP) of adults in the *B. bassiana* treatment group compared with the control group (*p* > 0.05). Days of spawning were 9.33 days for the LC_25_ treatment group, significantly lower (*p* < 0.05) when compared with the other two treatments. The average number of eggs laid by females was the lowest in the LC_25_ treatment group, with an average of only 133.02 eggs per female (*p* < 0.05). In addition, the male-to-female ratios of the offspring after the control and spray, LC_10_ and LC_25_, treatments were 1.16:1, 1.04:1 and 1:1, respectively.

The population parameters of the intrinsic growth rate (***r***) (0.143), the finite rate of increase (***λ***) (1.154) and the net reproductive rate (***R*_0_**) (61.69) of the LC_25_ treatment group were significantly lower than those of the LC_10_ treatment and the control groups (*p* < 0.05) (Table 4), indicating that the population growth rate of the LC_25_ treatment group decreased significantly, and the total number of offspring of individuals decreased. Compared with the control treatment (27.54 days), the average generation period ***T*** (28.73 days) of the LC_25_ treatment group was significantly prolonged (*p* < 0.05), indicating that the time required for the population to reach a stable growth rate to the ***R*_0_** became longer; there was no significant difference in various population parameters between the control and LC_10_ treatment groups (*p* < 0.05).

Figure 1 indicates the age-stage survival rate (***s_xj_***) of the offspring of WBPHs treated with LC_10_ and LC_25_ of *B. bassiana* and the control group. The results showed that the ***s_xj_*** curves of the F_1_ generation of the control and *B. bassiana*-treated groups had some overlap. The curves of pre-adult developmental stages had similar fluctuations among the three treatments, and the age-specific survival rate curves of adults all ended with male adults. The overall survival rate of nymphs and the age-stage survival rates of male and female adults in the LC_25_ treatment group were lower than those in the LC_10_ treatment and control groups, but the difference in survival rates between male and female adults was smaller.

Figure 2 indicates that the three population age-specific survival rate (***l_x_***) curves in the figure began to decline from the 29th to the 31st days, and the ***l_x_*** curve of LC_25_ decreased more rapidly, from 88% to 32% within 7 days. The female age-specific fecundity (***f_x_*_7_**) curve refers to the number of eggs laid by females per day at age ***x*** and stage ***7***. The graph showed that the control group reached a peak of 25.04 eggs on the 27th day. In contrast, the age-specific fecundity curves of LC_10_- and LC_25_-treated females reached the maximum on the 26th day (21.82 eggs) and 29th day (20.45 eggs), respectively. Age-specific fecundity of the total population (***m_x_***) refers to the average number of eggs laid by the entire population at age ***x***, and the ***m_x_*** of the LC_25_ treatment group was lower than that of the control and LC_10_ treatment groups. In addition, the age-specific maternity (***l_x_m_x_***) was the product of the population age-specific survival rate (***l_x_***) and the age-specific fecundity of the total population (***m_x_***), and the LC_25_-treated group in the figure had the lowest ***l_x_m_x_***, i.e., the total single female egg production was lower than other groups.

Figure 3 indicates the age-stage life expectancy curves of WBPHs under the three treatments. Age-stage life expectancy (***e_xj_***) refers to the time that individuals at age ***x*** and stage ***j*** can remain alive. The age-stage life expectancy (***e_xj_***) of the three treatments in the figure decreased with increasing age and stage. After being treated with *B. bassiana* at LC_25_ concentration, the life expectancy of newborn eggs was 33.8 days, which was lower than that of the control treatment (35.45 days). In addition, the age-stage life expectancy of male adults in the three treatments was higher than that of female adults.

The age-stage reproductive value (***v_xj_***) indicated the contribution of an individual at age ***x*** stage ***j*** to population growth. The results (Figure 4) showed that the age-stage reproductive value of the *B. bassiana* LC_25_-treated group was reduced compared to the control treatment group. The eclosion of females could increase the reproductive value, and the ***v_xj_*** of the female adults at the age stage of the three treatments had the highest peak on the 24th day, indicating that the females on the 24th day have a greater impact on population growth than other times, and the peak size was expressed as LC_10_ treatment (96.26 eggs/day) and control treatment (96.25 eggs/day) > LC_25_ treatment (73.52 eggs/day). In addition, the reproductive value curve of the 5^th^-instar nymphs in the LC_25_ treatment group appeared with two peaks, and the second peak was higher than the previous one.

## 4. Discussion

*B. bassiana* strain BEdy1 produces a fast growth rate, high sporulation, and high lethality rate to *E. d. yunnanus* adults [29]. Our previous preliminary study (unpublished) found that *B. bassiana* BEdy1 had higher pathogenicity to WBPHs than *B. bassiana* strain JZ21004 (Hubei Qiming Biological Company, Yidu City, China) used in production. For these reasons, *B. bassiana* BEdy1 has great prospects for application in the large-scale production and management of *S. furcifera*. However, when evaluating the control ability of entomogenous fungi against pests, the application potential of fungi is usually underestimated when only using the index of the lethal effect; actually, sublethal concentrations in fields occur when the concentration of insecticides gradually decreases after a preliminary application for a few days or weeks, and the host is very likely to be exposed to sublethal doses at that time [8,37]. Therefore, when *B. bassiana* is used to control WBPHs, it will produce a wide range of sublethal effects. Our goal is to explore the sublethal effect of *B. bassiana* on WBPHs by using the TWOSEX-MSChart Software [31]; it is important to assess the sublethal effects of *B. bassiana* on *S. furcifera* for the sustainability of agricultural production.

According to our results, when WBPH nymphs were treated with a sublethal dose of *B. bassiana*, which had adverse effects on the growth, development and reproduction of the parent and offspring, consistent with previous research results [38], the longevity of *Nilaparvata lugens* (Stål) (Homoptera: Delphacidae) parental females sublethal treated with *B. bassiana* NJBb2101 did not change the longevity significantly compared with the control, but the fecundity of F_0_ females decreased. The development duration of the offspring nymphs of WBPH treated with *B. bassiana* BEdy1 LC_25_ was significantly prolonged, and the average fecundity of F_1_ females was significantly lower than that of the control; similarly, the effect of entomopathogenic fungus on *N. lugens*, *Aedes albopictus* (Skuse) (Diptera: Culicidae), *Brevicoryne brassicae* (L.) (Hemiptera: Aphididae), *Bactrocera zonata* (Saunders) (Diptera: Tephritidae) and *Bactrocera dorsalisi* (Hendel) (Diptera: Tephritidae) was found to be that the nymphal stage was prolonged and fecundity was reduced compared with those of the control [38,39,40,41]. However, the difference is that neither *Metarhizium anisopliae* (Metschn.) Sorokīn (Hypocreales: Clavicipitaceae) nor *B. bassiana* had significant sublethal effects on the larval growth and oviposition of *Cyclocephala lurida* (Bland) (Coleoptera: Scarabaeidae) [42]. Contrarily, the larval period of *Spodoptera litura* (Fabricius) (Lepidoptera: Noctuidae) treated with *B. bassiana* was significantly decreased as compared to the control [43], some possible reasons for the difference were various *B. bassiana* strains and host species. In addition, some studies have shown that entomogenous fungi can deter pests from feeding, resisting immune defense and using host nutrients [44,45,46], these factors may cause the decline of the fitness of *S. furcifera*.

The assessment of life table parameters is essential for determining the overall sublethal effects of entomogenous fungi on insect survival, developmental time, and reproduction. Through data analysis of life table parameters, the ***r***, ***λ***, ***R*_0_** and ***T*** of the F_1_ generation changed significantly, and the ***r***, ***λ*** and ***R*_0_** of the LC_25_ treatment group were significantly shorter than those of the control group. Similarly, the ***r***, ***λ*** and ***R*_0_** were shorter than the control but ***T*** was not changed significantly after *Bactericera cockerelli* (Sulc) (Hemiptera: Triozidae) were treated with sub-lethal doses of *B. bassiana* [47]. Our results showed that the ***T*** of the LC_25_ treatment was significantly longer than that of the control; similarly, *B. bassiana* had this effect on *Helicoverpa armigera* (Hübner) (Lepidoptera: Noctuidae) and *Frankliniella occidentalis* (Pergande) (Thysanoptera: Thripidae) [48,49]. Our data indicated that *B. bassiana* LC_25_ may have a long-term influence on *S. furcifera* physiology and an inhibitory effect on the population growth of WBPH.

In addition, the survival rate (***s_xj_*** and ***l_x_***) can reflect the population growth of insects after fungal treatment, which is an important reference for judging the adaptability of insect populations to pathogenic fungi [50]. The population growth of potato tuber moth was inhibited after being treated with *B. bassiana*, which showed that the age-stage survival rate (***s_xj_***) of each age stage was lower than that of the control, and the age-specific survival rate (***l_x_***) of the population showed a linear downward trend [51]. Additionally, in this study, the ***s_xj_*** of male adults of the LC_25_ treatment group was always higher than the ***s_xj_*** of female adults, which was unfavorable to the development of the population, and the ***l_x_*** of the treatment group also decreased faster than the control, indicating that the population development of the offspring was inhibited. According to ***v_xj_*** and ***e_xj_***, the reproductive values of female adults in the *B. bassiana* treatment group decreased, and the life expectancy was lower than that of the control, which was consistent with the results of *A. albopictus* and *F. occidentalis* [39,49], which showed that the offspring population of the *S. furcifera* had poor adaptability to *B. bassiana* at LC_25_ concentration.

The purpose of the current study was to determine the sublethal effects of *B. bassiana* BEdy1 on *S. furcifera* and this study has shown that sublethal dose *B. bassiana* BEdy1 can affect the host growth, development and reproduction, as well as inhibiting the population growth, of WBPHs, and the results justified the significance of assessing the sublethal effects of *B. bassiana* on *S. furcifera* populations in the field. These observations also have important implications for the long-term management of *S. furcifera*. However, a limitation exists in our study: as with other studies on the sublethal effects of the WBPH [24,25,26,27,28], the experiment was completely carried out under laboratory conditions, and many additional elements may influence the dynamics of WBPH populations, such as excessive nitrogen fertilization, geographic location and weather variables [52,53,54]. In order to comprehensively analyze the growth and development parameters of the entire population, these experimental conditions should be combined with the actual conditions in the field. Furthermore, the effect of *B. bassiana* on insects is the result of multiple factors, such as mechanical damage resulting from tissue destruction, depletion of nutrient resources and toxicosis [55], and further work is needed to reveal the interaction between *B. bassiana* and *S. furcifera*; an in-depth study of these possible theories will be of some help in improving the IPM of rice.

## Figures and Tables

**Figure 1 jof-09-00123-f001:**
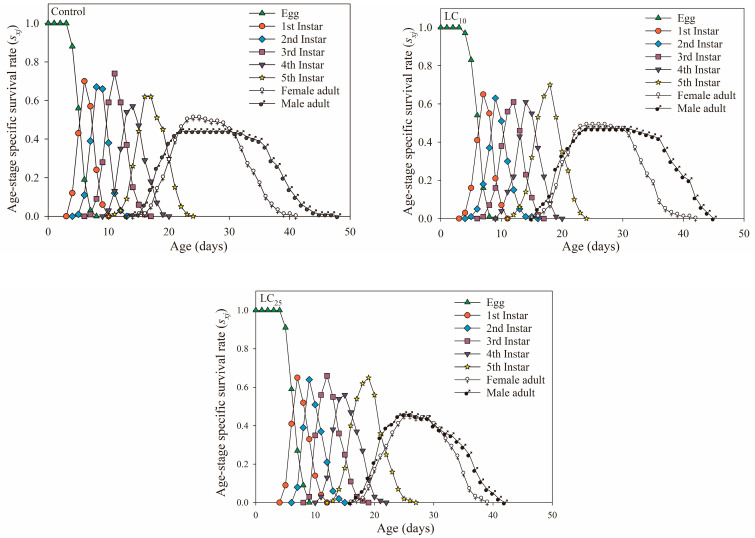
Age-stage-specific survival rates (***s_xj_***) in the control, *Beauveria bassiana* LC_10_- and LC_25_-treated F_1_ generation of *Sogatella furcifera*.

**Figure 2 jof-09-00123-f002:**
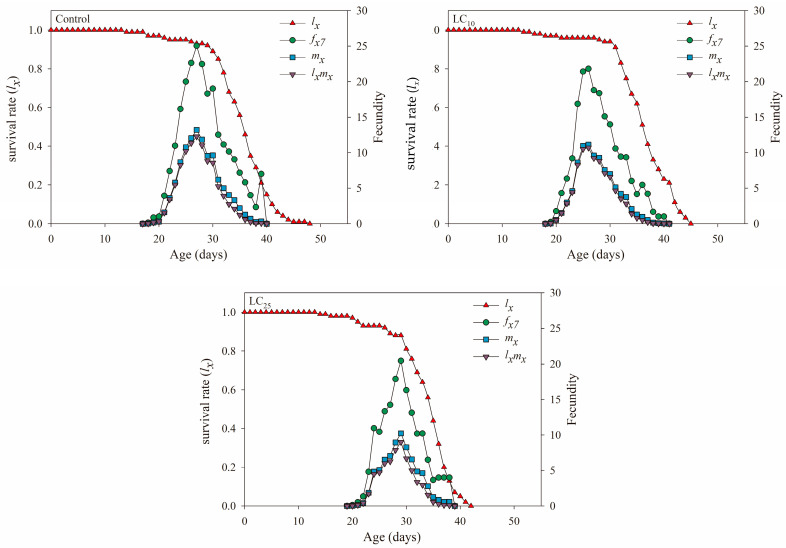
Age-specific survival rate (***l_x_***), female age-specific fecundity (***f_x_*_7_**), age-specific fecundity of the total population (***m_x_***) and age-specific maternity (***l_x_m_x_***) in the control, *Beauveria bassiana* LC_10_- and LC_25_-treated F_1_ generation of *Sogatella furcifera*.

**Figure 3 jof-09-00123-f003:**
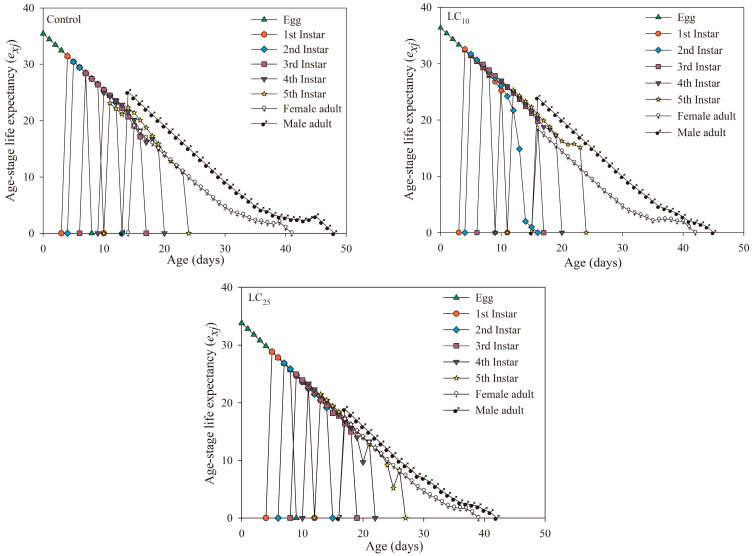
Age-stage specific life expectancy (***e_xj_***) in the control, *Beauveria bassiana* LC_10_- and LC_25_-treated F_1_ generation of *Sogatella furcifera*.

**Figure 4 jof-09-00123-f004:**
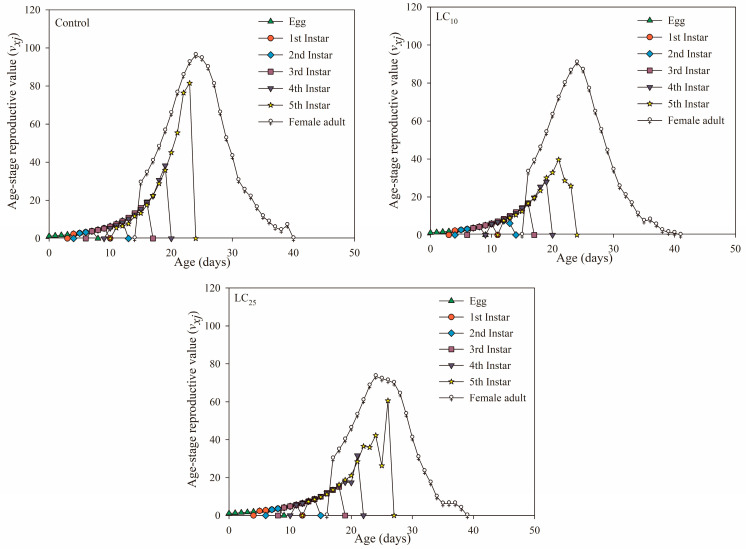
Age-stage reproductive value (***v_xj_***) in the control, *Beauveria bassiana* LC_10_- and LC_25_-treated F_1_ generation of *Sogatella furcifera*.

**Table 1 jof-09-00123-t001:** The sublethal concentrations of *Beauveria bassiana* BEdy1 to the 3rd-instar nymphs of *Sogatella furcifera*.

Strain	N	LC_10_ with 95% CI (Spores/mL) ^a^	LC_25_ with 95% CI (Spores/mL)	*χ*^2^ (*df*) ^b^
BEdy1	540	8.29 × 10^3^ (1.41 × 10^2^, 6.91 × 10^4^)	1.65 × 10^6^ (2.99 × 10^5^, 5.14 × 10^6^)	2.834 (13)

^a^ 95% confidence limits. ^b^ Chi-square value (*χ*^2^) and degrees of freedom (*df*) as calculated using probit analysis (Polo Plus 2.0).

**Table 2 jof-09-00123-t002:** The effects of sublethal concentrations of *Beauveria bassiana* BEdy1 on adult longevity and fecundity of the F_0_ generation *Sogatella furcifera*.

Parameters	Control	LC_10_	LC_25_
Longevity (days)	16.27 ± 0.35a	15.85 ± 0.32a	15.31 ± 0.34a
Mean fecundity (eggs/female)	156.29 ± 7.18a	167.82 ± 4.59a	131.04 ± 7.12b

Values are means ± SEs (standard errors of the means). The different letters in the same line indicate significant differences (*p* < 0.05).

**Table 3 jof-09-00123-t003:** The effects of sublethal concentrations of *Beauveria bassiana* BEdy1 on the development duration, longevity and reproductive parameters of the F_1_ generation of *Sogatella furcifera*.

Stage or Parameters	Control	LC_10_	LC_25_
Egg (d)	5.66 ± 0.10c(100)	6.51 ± 0.10b(100)	6.86 ± 0.11a(100)
1st-instar nymph (d)	2.12 ± 0.05a(100)	2.08 ± 0.04a(100)	2.18 ± 0.04a(100)
2nd-instar nymph (d)	2.37 ± 0.05a(100)	2.20 ± 0.05b(98)	2.28 ± 0.06ab(100)
3rd-instar nymph (d)	2.89 ± 0.08a(100)	2.66 ± 0.07b(98)	2.88 ± 0.10ab(98)
4th-instar nymph (d)	2.58 ± 0.08bc(98)	2.70 ± 0.07b(97)	2.95 ± 0.10a(97)
5th-instar nymph (d)	3.60 ± 0.12b(95)	3.76 ± 0.10ab(96)	3.97 ± 0.10a(92)
Pre-adult (d)	19.25 ± 0.21c(95)	19.92 ± 0.20b(96)	21.11 ± 0.22a(92)
Adult longevity (d)	17.08 ± 0.43a(95)	17.23 ± 0.40a(96)	13.87 ± 0.35b(92)
Female adult longevity (d)	13.98 ± 0.33a(51)	14.22 ± 0.36a(49)	12.43 ± 0.41b(46)
Male adult longevity (d)	20.68 ± 0.41a(44)	20.36 ± 0.36a(47)	15.30 ± 0.48b(46)
APOP (d)	2.90 ± 0.08ab(50)	2.89 ± 0.06b(46)	3.10 ± 0.06a(42)
TPOP (d)	22.92 ± 0.29b(50)	23.00 ± 0.26b(46)	24.76 ± 0.35a(42)
Oviposition days (d)	10.52 ± 0.32a(50)	10.57 ± 0.29a(46)	9.33 ± 0.34b(42)
Fecundity (offspring/individual)	191.08 ± 7.61a(51)	172.08 ± 7.97a(49)	133.02 ± 8.23b(46)

The different letters in the same line indicate significant differences (*p* < 0.05). The numbers in parentheses represent the number of tested insects at a particular stage. The (d) represents days. APOP: adult preoviposition period, TPOP: total preoviposition period.

**Table 4 jof-09-00123-t004:** The effects of sublethal concentrations of *Beauveria bassiana* BEdy1 on population parameters of F_1_ generation of *Sogatella furcifera*.

Population Parameters	Control	LC_10_	LC_25_
Intrinsic rate of increase, ***r*** (d^−1^)	0.166 ± 0.004a	0.162 ± 0.005a	0.143 ± 0.005b
Finite rate of increase, ***λ*** (d^−1^)	1.181 ± 0.005a	1.179 ± 0.005a	1.154 ± 0.006b
Net reproductive rate, ***R*_0_** (eggs/individual)	97.45 ± 10.30a	84.32 ± 9.47ab	61.19 ± 7.64b
Mean generation time, ***T*** (d)	27.544 ± 0.313b	27.415 ± 0.290b	28.733 ± 0.307a

Values are means ± SEs (standard errors of the means). The different letters in the same line indicate significant differences (*p* < 0.05).

## Data Availability

Not applicable.

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
