# Peer review of "Sublethal Effects of Beauveria bassiana Strain BEdy1 on the Development and Reproduction of the White-Backed Planthopper, Sogatella furcifera (Horváth) (Hemiptera: Delphacidae)"

_jof, 2023, doi:10.3390/jof9010123_

Round 1

Reviewer 1 Report

very good

Author Response

                                  Response to Reviewer 1 Comments

Comment 1: very good.

Response: We thank the reviewer for reading our paper carefully and giving the above positive comments.

Reviewer 2 Report

The authors have carried out an interesting study whose main objective was to evaluate the sublethal effects of Beauveria bassiana on the white-backed planthopper, a serious pest of rice in Asia.  In my opinion the work is well written, the methodology is adequately presented and the results are consistent. I consider that it could be published in its current form. However, I suggest authors taking into account some observations that I have made throughout the manuscript and that I mention below. 

Abstract

Line 14: White-backed Planhopper should be previously named in the summary to be able to mention the abbreviation of it later.

Introduction

Line 52: Please add (Bals.-Criv.) Vuill. (Hypocreales:Cordycipitaceae) after Beauveria bassiana

Line 69: Please add Heller (Coleoptera: Curculionidae) after Ergania doriae yunnanus

Materials and Methods

Line 99: Please clarify if the nymphs had recently moulted and how they were monitored. This information is important to prevent the conidia from detaching together with the moult and failing to penetrate the integument of the host.

Discussion

Line 285: Please replace Metarhizium anisoplia by Metarhizium anisopliae and add (Metschn.) Sorokīn (Hypocreales: Clavicipitaceae) after the specific name.

Line 314: Add “and” after Aedes albopictus.

Author Response

Dear reviewer,

Thanks for your comments. Those comments: are all valuable and helpful for revising and improving our paper. The main corrections in the paper and the responds to the reviewer' comments are as flowing:

Comment 1: Line 14: White-backed Planhopper should be previously named in the summary to be able to mention the abbreviation of it later.

Response: Thanks for your suggestion. We have previously named “white-backed planthopper (WBPH)” in Line 11 according to the reviewer’s comments.

Comment 2: Line 52: Please add (Bals.-Criv.) Vuill. (Hypocreales:Cordycipitaceae) after Beauveria bassiana.

Response: Thanks for your kind suggestions. We added (Bals.-Criv.) Vuill. (Hypocreales:Cordycipitaceae) after Beauveria bassiana in Line 14 and Line 55.

Comment 3: Line 69: Please add Heller (Coleoptera: Curculionidae) after Ergania doriae yunnanus.

Response: Thanks for your kind suggestions. We added Heller (Coleoptera: Curculionidae) after Ergania doriae yunnanus in Line 72.

Comment 4: Line 99: Please clarify if the nymphs had recently moulted and how they were monitored. This information is important to prevent the conidia from detaching together with the moult and failing to penetrate the integument of the host.

Response: We thank the reviewer for pointing out this issue. We collected healthy and uniform 3rd-instar nymphs for bioassay by observing them under the microscope every day, and we clarified in Line 102.

Comment 5: Line 285: Please replace Metarhizium anisoplia by Metarhizium anisopliae and add (Metschn.) Sorokīn (Hypocreales: Clavicipitaceae) after the specific name.

Response: Thanks for your kind suggestions. We have replaced Metarhizium anisoplia by Metarhizium anisopliae and add (Metschn.) Sorokīn (Hypocreales: Clavicipitaceae) after the specific name in Line 305.

Comment 6: Line 314: Add “and” after Aedes albopictus.

Response: Thanks for your kind suggestions. We added “and” after Aedes albopictus in Line 339.

We tried our best to improve the manuscript and made some changes in the manuscript, and we deeply appreciate your consideration and comments of our manuscript. Looking forward to hearing from you.

Thank you and best regards.

Yours sincerely,

                                                             Mr. Yongbo Xia

                                                 Corresponding author: Qunfang Yang

                                                                     2023.1.8

Reviewer 3 Report

Comments and suggestions are found in the document. 

Comments and suggestions are found in the document.

But the most important thing is:

How these results may have an impact under field conditions, in relation to the damage caused by S. furcifera and the transmission of the virus (Southern rice black-streaked dwarf virus (SRBSDV)). Since the population of S. furcifera continues to grow slowly but continues to grow.

What does B. bassiana do to S. furcifera population dynamics?

How might this affect the damage caused by S. furcifera?

Author Response

Dear reviewer,

Thanks for your comments. Those comments are all valuable and helpful for revising and improving our manuscript. The main corrections in the manuscript and the responds to the reviewer' comments are as flowing :

Comment 1: (WBPHs) These are the initials of the insect's common name, but it is not specified at the beginning of the insect's scientific name. Include the common name of the insect when mentioned for the first time in the abstract, as it was mentioned in the introduction.

Response: Thanks for your kind suggestions. We have mentioned “white-backed planthopper (WBPH)” in the abstract in Line 11 according to the reviewer’s comments.

Comment 2: Preferably include keywords that are not repeated in the title of the manuscript.

Response: Thanks for your comments. According to the revised content, we have replaced Beauveria bassiana by entomogenous fungi in Line 27.

Comment 3: decreased activity, What activity.

Response: Thanks for your comments. We wanted to express rice plants decreased activity (such as photosynthesis) after the leaves turn yellow, we think use “reduce plant vigor” describe more accurately, so we have replaced” decreased activity” by “reduce plant vigor” in Line 38.

Comment 4:Why 3rd-instar nymphs? At what stage does it begin to feed and transmit virus? it's important to know?

Response: Thanks for your comments. Due to stable growth and development of 3rd-instar nymphs, reliable results can be obtained by bioassay of 3rd-instar nymphs. We feed them all life cycle Both adults and nymphs can transmit the virus. The virus is not involved in our research, so we did not feed the virus, so we do not think it's important to know in our manuscript.

Comment 5:It is not clear. Each treatment (concentration) was spread on 30 third-instar nymphs? Or All concentrations were sprayed on 30 third-instar nymphs? Good to clarify.

Response: Thanks for your comments. We collected 30 healthy and uniform 3rd-instar nymphs with a suction trap as one group. For each group 3 ml of the spore suspension was sprayed on the body wall of 30 3rd-instar nymphs, we have clarified it in Line 104.

Comment 6: Exactly 3 ml or 3±SD ml. How do you know it was exactly 3 ml? The methodology must be replicated by other researchers.

Response: Thanks for your comments. We use a pipette(1000μl) to extract 3ml of spore suspension into a hand-held spray for bioassay, other researchers are able to replicate bioassay.

Comment 7: Add cite.

Response: We are very sorry for our incorrect writing. We added the cite in Line 157.

Comment 8: The results showed that F1 generation of WBPH treated with LC10 and LC25 of B. bassiana could complete its development.

Response: Thanks for your comments. We used” The results showed that F1 generation of WBPH treated with LC10 and LC25 of B. bassiana could complete its development.” As first sentence in Line 178, according to the suggestions of reviewers.

Comment 9: It is not clear. Good to clarify. Please replace "egg stages" with "development periods of eggs." nymph stage with development times of the 1st-instar nymph SINCE this the results are reported in days Review entire subsection3.3 Rewrite the sentence.

Response: Thanks for your comments. We have replaced "egg stages" with "development periods of eggs." nymph stage with development times of the 1st-instar nymph 2nd-instar nymph, 3rd-instar nymph, 4th-instar nymph, 5th-instar nymph and pre-adult in subsection3.3.

Comment 10: These results do not appear in any table.

Response: Thanks for your comments. We have replaced “longevity(F)” by “Female adult longevity” and replaced “longevity(M)” by “Male adult longevity”, because the numbers in parentheses represent the number of tested insects at a particular stage, so we know the number of male and female adults, and calculated ratios in Line 200.

Comment 11: Meaning of (d). Include at footnote of the table.

Response: Thanks for your comments. We have explained the meaning of (d), APOD and TPOD at footnote of the table in Line 205.

Comment 12: three treatments?

Response: We are very sorry for our incorrect writing. We added the “treatments” after “three” in Line 227.

Comment 13: What is the statistical method used by the Software to compare the curves? It was not clarified in the materials and methods section.

Response: We thank the reviewer for pointing out this issue. All curves plotted using Sigmaplot 14.0 (Systat Software, Inc.). The significant difference between parameters was calculated by the paired bootstrap test in TWOSEX-MSChart Software in (Line 157). We found it is inappropriate to description curves in “significantly”, and we deleted “significantly” in Line 226, Line 241, Line 245, Line 255, Line 257 and Line 265, respectively.

Comment 14: italics?

Response: We are very sorry for our incorrect writing. We have replaced “strain” by “strain”.

Comment 15: Add taxonomic authority, order and family.

Response: Thanks for your comments. We added taxonomic authority, order and family after Nilaparvata lugens, Aedes albopictus, Brevicoryne brassicae,

Bactrocera zonata, Bactrocera dorsalisi, Cyclocephala lurida, Spodoptera litura, respectively, in Line 295, Line 301, Line 302, Line 305, Line 307 and 308. In addition, we added taxonomic authority, order and family after Bactericera cockerelli, Helicoverpa armigera and Frankliniella occidentalis Line 319, Line 322 and Line 323.

Comment 16: How these results may have an impact under field conditions, in relation to the damage caused by S. furcifera and the transmission of the virus (Southern rice black-streaked dwarf virus (SRBSDV)). Since the population of S. furcifera continues to grow slowly but continues to grow. What does B. bassiana do to S. furcifera population dynamics? How might this affect the damage cause by S. furcifera?

Response: Thanks for your comments. It is really true as reviewer commented that in relation to the damage caused by S. furcifera and the transmission of the virus may have an impact under field conditions. Due to we didn't think enough, the factor of Southern rice black streaked dwarf virus was not taken into account when designing the experiment, in our study, the purpose was to determine relationship entomogenous fungi (Beauveria bassiana Strain BEdy1) between host insects (Sogatella furcifera). In the future, we need to verify what B. bassiana do to S. furcifera population dynamics and how might this affect the damage cause by S. furcifera under complex conditions in the field.

We tried our best to improve the manuscript and made some changes in the manuscript, and we deeply appreciate your consideration and comments of our manuscript. Looking forward to hearing from you.

Thank you and best regards.

Yours sincerely,

                                                              Mr. Yongbo Xia

                                                  Corresponding author: Qunfang Yang

                                                                     2023.1.8

Reviewer 4 Report

Detailed comments are attached 

Author Response

Dear reviewer,

Thanks for your comments. Those comments are all valuable and helpful for revising and improving our manuscript. The main corrections in the manuscript and the responds to the reviewer' comments are as flowing :

Comment 1: In the abstract section the of the words like” WBPHs and TPOP” should be avoided.

Response: Thanks for your suggestion. We have previously named “white-backed planthopper (WBPH)” in Line 11 and each short form has an explanation.

Comment 2: Keywords should not be similar as of the title words.

Response: Thanks for your kind suggestions. We have replaced Beauveria bassiana by entomogenous fungi in Line 27.

Comment 3: Authority should be mentioned with scientific names at first mentioned “Rice (Oryza sativa)”.

Response: Thanks for your kind suggestion. We added “(Oryza sativa)” after "Rice" in Line 11.

Comment 4: The reference numbers insertion is different throughout the manuscript such as in the introduction section “[1,2]. [3.4.5]. [6–8]. [3.23.24.25.26.27]”. Plz. Check it. In the introduction section plz. cite latest work regarding as insect control; like, Insects 2022, 13, 103. https://doi.org/10.3390/insects13010103

Response: Thank you for pointing out this problem in our manuscript. We checked all references, we have replaced [3–5] by [3.4.5] in Line 37, and replaced [3.24–28] by [3.23.24.25.26.27] in Line 69

Thanks for the references, which are now included in the revised manuscript in Line 16.

Comment 5: When a paragraph is started with the scientific name, the scientific name should be written completely. “S. furcifera” e.g, under subheading 2.1.

Response: Thank you for pointing out this problem in our manuscript. We have replaced B. bassiana by Beauveria bassiana in Line 72 and 91, and replaced S. furcifera by Sogatella furcifera in Line 82.

Comment 6: F1 generation or F1 generation? be consistent.

Response: Thank you for pointing out this problem in our manuscript. F1 should be used, we have modified manuscript and it has been consistent.

Comment 7: Figure 1. Legend should have sufficient information to explain the figures thoroughly. Each graph should represent the LC25, LC10 and control for clear comparison and better understanding. Author should separate the graphs. In the present graphs it is difficult for the reader to compare and understand the results. It is recommended for all figures.

Response: Thanks for your comments. According to the suggestions of reviewer, in order to clear comparison and better understanding each graph we used “Age-stage-specific survival rates (sxj) in the control, Beauveria bassiana LC10 and LC25 treated F1 generation of Sogatella furcifera.” in Figure 1(Line 230),used “Age-specific survival rate (lx), female age-specific fecundity (fx7), age-specific fecundity of the total population (mx) and age-specific maternity (lxmx) in the control, Beauveria bassiana LC10 and LC25 treated F1 generation of Sogatella furcifera.” in Figure 2(Line 247), used “Age-stage specific life expectancy (exj) in the control, Beauveria bassiana LC10 and LC25 treated F1 generation of Sogatella furcifera” in Figure 3(Line 259) and used” Age-stage reproductive value (vxj) in the control, Beauveria bassiana LC10 and LC25 treated F1 generation of Sogatella furcifera” in Figure 4(Line 274).

In order to intuitively and clearly compare the differences between the same curves of different treatments, we think it is unnecessary to separate the graphs.

Comment 8: Write complete scientific names of the organisms in the figure legends.

Response: Thanks for your comments. We write complete scientific names of the organisms in Figure 1(Line 230), Figure 2(Line 249), Figure 3(Line 260) and Figure 4(Line 275). Besides, we write complete scientific names of the organisms in Table 1(Line 163), Table 2(Line 173), Table 3(Line 201) and Table 4(Line 216).

Comment 9: There is no conclusions section.

Response: Thanks for your comments. We have come to the conclusion in the last paragraph of the discussion, so we did not write the conclusions section. According to the Journal of Fungi, this section is not mandatory.

We tried our best to improve the manuscript and made some changes in the manuscript, and we deeply appreciate your consideration and comments of our manuscript. Looking forward to hearing from you.

Thank you and best regards.

Yours sincerely,

                                                    Mr. Yongbo Xia

                             Corresponding author: Qunfang Yang

                                                         2023.1.8

Round 2

Reviewer 4 Report

The author did most of the changes but major suggestion has not been followed. This point is given below. Moreover, the conclusion section is not added. 

Figure 1. Legend should have sufficient information to explain the figures thoroughly. Each graph should represent the LC25, LC10 and control for clear comparison and better understanding. Author should separate the graphs. In the present graphs it is difficult for the reader to compare and understand the results. It is recommended for all figures.